

# The non-indigenous dung beetle (*Onthophagus nuchicornis*) can effectively reproduce using the dung of indigenous eastern North American mammals

Alexe Indigo[1], Katelyn Stokes[1], Olivia Burchell[2] and Paul Manning[1]

[1] Department of Plant, Food and Environmental Sciences, Faculty of Agriculture, Dalhousie University, Truro, Nova Scotia, Canada
[2] Department of Animal Science and Aquaculture, Faculty of Agriculture, Dalhousie University, Truro, Nova Scotia, Canada

## ABSTRACT

Non-indigenous dung beetle (Coleoptera: Scarabaeoidea) species in North America are important contributors to ecosystem functions, particularly in pasture-based livestock systems. Despite the significant body of research surrounding non-indigenous (and often invasive) dung beetles in agricultural contexts, there has been minimal study concerning the impact that these species may have on indigenous dung beetle populations in natural environments. Here we examine the possible impact of the introduced dung beetle *Onthophagus nuchicornis* on indigenous dung beetle populations via use of indigenous mammal dung. Using a controlled laboratory experiment, we quantified how readily beetles could use dung from bobcat (*Lynx rufus*), red fox (*Vulpes vulpes*), moose (*Alces alces*), raccoon (*Procyon lotor*), and domestic sheep (*Ovis aries*). To determine the suitability of each dung type for reproduction, we measured offspring abundance and fitness. While the number of developed offspring was significantly different among treatments, offspring fitness did not vary significantly across dung types. Our results demonstrate the generalist feeding habits of the non-indigenous dung beetle *O. nuchicornis* can allow this species to use the dung of various wild animals indigenous to eastern North America.

## INTRODUCTION

Non-indigenous insect species can be introduced to a new environment through deliberate or accidental means. If a species establishes successfully, an introduction has the potential to create negative impacts for indigenous species and ecosystems in various ways, including predation, parasitism, and competition with indigenous species for resources (*e.g.*, nesting sites, food) (*Parker et al., 1999*; *Langor et al., 2014*). There may also be broader impacts to the surrounding ecosystem, such as changes to resource and nutrient dynamics affecting the organisms and environment linked to the same community (*Parker et al., 1999*; *Pokhrel, Cairns & Andrew, 2020*).

Corresponding author
Alexe Indigo, alexe.indigo@dal.ca

In some circumstances, introduced insect species can provide environmental benefits (*Pokhrel et al., 2021*). Dung beetles (Coleoptera: Scarabaeoidea), for example, provide critical ecosystem functions in pasture-based livestock systems (*Nichols et al., 2008*). Through their consumptive and reproductive activities, dung beetles contribute to the dispersal and burial of dung, which improves pasture quality, recycles nutrients back into the soil, mitigates greenhouse gas production, and can reduce livestock pest and parasite populations (*Nichols et al., 2008*). Dung beetles have been deliberately introduced in North America by numerous government-sponsored programs throughout the 1900s, primarily for the purpose of improving degradation of cattle dung. Alongside deliberate introductions, some dung beetle species have been unintentionally introduced during European colonization on boats carrying livestock or that used soil as ballast (*Hoebeke & Beucke, 1997*).

Some introduced dung beetle species, once naturalized, have become invasive by spreading beyond their intended range (*Montes de Oca & Halffter, 1998*; *MacRae & Penn, 2001*; *Filho et al., 2018*). Invasive dung beetles exhibit many characteristics typical of invasive species, including high lifetime fecundity, multiple generations per year, high dispersal, and a generalist diet—all of which are demonstrated by the most successfully invasive dung beetle species worldwide (as determined by *Pokhrel, Cairns & Andrew, 2020*). Though environmental filters are thought to limit species adapted to pasture environments from colonizing less suitable habitats, behavioural and physiological plasticity of dung beetles can facilitate colonization. This is epitomized by the case of *Digitonthophagus gazella*, an Afro-Asian dung beetle associated with pasture and savannah that has been sampled in a variety of Brazilian forests following an intentional introduction effort (*Filho et al., 2018*).

One species of dung beetle that has successfully invaded a large geographical area beyond its original range is *Onthophagus nuchicornis* (L. 1758). *Onthophagus nuchicornis* is a Palearctic dung beetle species believed to have been unintentionally introduced to North America in the 1840s (*Hoebeke & Beucke, 1997*). Today, *O. nuchicornis* has been observed in all ten Canadian provinces and throughout the northern United States (*Hoebeke & Beucke, 1997*). Like other successfully invasive dung beetle species, it is a generalist feeder and is abundant in cattle-grazed pasture throughout its North American range. In its native range, *O. nuchicornis* is associated with cattle, sheep, horse, dog, and rabbit dung (*Biström, Silfverberg & Rutanen, 1991*; *Mann & Lane, 2016*). *Onthophagus nuchicornis* has an obligatory cold reproductive diapause (*Floate et al., 2015*); while this prevents *O. nuchicornis* from producing multiple generations per year (a characteristic often observed in invasive insects), it has allowed *O. nuchicornis* to colonize regions of North America that have winters too cold for dung beetle species adapted to warmer climates (*Floate et al., 2015*). Studies have also shown that *Onthophagus* species, though predominantly observed in cattle pastures, have also been frequently observed in woodland environments and are attracted to a broad range of dung sources (*Fincher, Stewart & Davis, 1970*; *Floate, 2023*), indicating the potential for this species to spread into non-agricultural habitats.

While introduced dung beetle species in North America are often important contributors to ecosystem functions in managed agricultural environments, they also have the potential to outcompete indigenous species, negatively impacting insect biodiversity. Despite significant research concerning introduced (and often invasive) dung beetles in agricultural contexts, there has been minimal study concerning the impact that introduced dung beetles may have on indigenous dung beetle species, such as *Onthophagus hecate*, in non-managed environments. The congeneric species *Onthophagus hecate* is one of the most abundant species of dung beetle in its indigenous range of North America (*Ratcliffe & Paulsen, 2008*), though in our own observations is found far less frequently than *O. nuchicornis*. Given that *O. nuchicornis* has a generalist diet, is abundant in agricultural environments, and thrives in northern North American climates, the question of its potential impact on insect biodiversity beyond the agricultural ecosystem is highly relevant.

Determining the impact that *O. nuchicornis* may have on indigenous dung beetle populations in non-managed environments begins with understanding if *O. nuchicornis* can effectively use the dung of wild mammals indigenous to North America. In this study we investigate the ability of the non-native dung beetle *O. nuchicornis* to successfully reproduce using dung from various North American mammals, representing herbivorous, omnivorous, and carnivorous feeding guilds. Specifically we explore the following research questions using sheep dung as a baseline preferred resource:

(1) Do adult *O. nuchicornis* vary in how long they remain resident in the dung of wild mammals?
(2) Can *O. nuchicornis* effectively use the dung of wild animals for reproduction?
(3) Does variability in dung environment and nutrition correspond to fitness differences in the development of dung beetles?

## MATERIALS & METHODS

### Dung collection

Dung was collected from animals residing at the Shubenacadie Provincial Wildlife Park in Shubenacadie, Nova Scotia (45.09228, −63.39515) between 3–16 May 2023 (Fig. 1). Shubenacadie Provincial Wildlife Park houses a variety of indigenous and non-indigenous species that have either been born in captivity or are wild animals rehabilitated and deemed unsuitable for release. Four resident species indigenous to eastern North America were selected for inclusion in this study: bobcat (*Lynx rufus*), red fox (*Vulpes vulpes*), moose (*Alces alces*), and raccoon (*Procyon lotor*). These species were chosen to represent carnivorous (bobcat), omnivorous (red fox and raccoon), and herbivorous (moose) feeding guilds. A minimum of 350 g of fresh dung (deposited within the previous 18 h and no apparent previous dung beetle use) was collected from each species. Domestic sheep (*Ovis aries*) dung was also collected from a pasture located on the Dalhousie University Faculty of Agriculture's campus farm (45.37267, −63.25747). Following collection, dung samples were examined, and any older fragments or debris were removed. The dung was then stored at −18 °C until use.

Due to documented non-target effects of parasiticide residues on coprophagous insects (*e.g.*, *Floate et al., 2005*), no animal from which dung was collected had been treated within

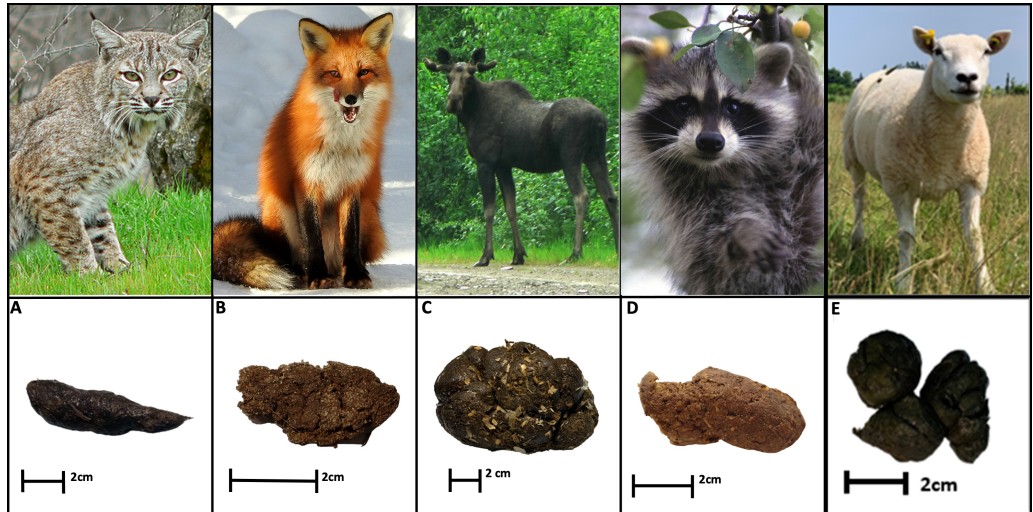

**Figure 1** **Fresh dung of four species indigenous to eastern North America, and domestic sheep.** (A) Bobcat. (B) Fox. (C) Moose. (D) Raccoon. (E) Texel Sheep. Image credits: Photo of the sheep was taken by Paul Manning; animal dung photos were taken by Katelyn Stokes. All other photos are in the public domain, licensed under the Creative Commons CC0 license, and were retrieved from Wikimedia Commons.

the 12 months prior to collection. The date and product of each species' last deworming, as well as the number of animals housed per enclosure was recorded (Table S1).

Dietary information for each of the species was obtained and compared to a typical wild diet for that species to estimate dietary alignment; all animals from which dung was collected received a diet to meet their nutritional requirements (Table 1).

A homogenized sample of each dung treatment was sent to the Nova Scotia Department of Agriculture's Analytic Laboratory in Bible Hill, Nova Scotia to determine the nutritional composition of each dung treatment (Table 1). Samples were thoroughly homogenized using a drill with a paint mixer attachment. A standardized feed analysis test provided the dry matter (DM; 100-moisture (AOAC, 2005; method no. 934.01)), crude protein (AOAC, 2005; method no. 990.03), crude fat (AOAC, 2005; method AM 5-04), carbon-to-nitrogen ration (LECO, 2024), and mineral content (AOAC, 2003; method no. 968.08) for each sample.

## Beetle collection

Beetles used in this study were collected from two sheep pastures between 6–14 July 2023 in Brookside, Nova Scotia (45.38817, −63.25576) and Bible Hill, Nova Scotia (45.37193, -63.25564). Sheep dung was searched by hand for *O. nuchicornis* adults. Male and female beetles were isolated according to sex (determined by the presence or absence of a cephalic horn) for the duration of the collection period (6-14 July 2023) in opaque polypropylene holding containers with damp paper towel until the experiment began. Holding containers had drainage holes in the bottom and a mesh lid for ventilation. An initial 95 g of fresh sheep dung was provided to each container during the holding period, with freshly wetted paper towel and 50 g fresh dung provided every two days.

**Table 1 Comparison of captive versus wild diets and dung nutritional composition for four eastern North American species and the domestic sheep from which dung was collected.** The mass range of a single deposition of dung was estimated for each species. Nutritional compositions were determined through a feed analysis test performed by the Nova Scotia Department of Agriculture's Analytic Lab.

| | Date collected (2024) | Captive diet | Wild diet | Captive and wild diet alignment | Estimated dung size (g) | % Dry matter | % Crude protein | % N | C:N |
|---|---|---|---|---|---|---|---|---|---|
| Bobcat | 4–6 May | Rabbit, dry cat food, chicken necks, red meat pieces and Toronto Zoo Carnivore Diet (primarily ground horse meat). | Lagomorphs (hare, rabbit), squirrel, porcupine, grouse, deer (*Christopherson et al., 2019*) | Moderate | 40-80 | 41.07 | 4.97 | 0.8 | 6.92 |
| Fox | 4–16 May | Rabbit, dry dog food, chicken necks, red meat pieces, Toronto Zoo Carnivore Diet, and boiled eggs | Small rodents (mice, voles, shrews), hare, fruit (*Morin et al., 2020*) | Moderate | 30–70 | 68.42 | 4.78 | 0.73 | 8.36 |
| Moose | 3-6 May | Mazuri Moose Pellets, pasture | Willow, alder, birch twigs and leaves (*Whiteside, 2009*) | Moderate | 200–1000 | 24.62 | 2.56 | 0.37 | 30.0 |
| Racoon | 4-6 May | Dry dog food, carrots, chickpeas, apple, banana, Toronto Zoo Carnivore Diet, and boiled eggs | Fruits, grasses, nuts, invertebrates, small vertebrates, crustaceans, eggs (*Dibello, Arthur & Krohn, 1990*) | Low | 30–80 | 22.32 | 5.71 | 0.89 | 10.0 |
| Sheep | 7–12 July | Pasture (grazing) | Pasture (grazing) | Very High | 100–600 | 26.37 | 4.30 | 0.67 | 18.6 |

## Experimental design

This study used $N = 20$ mesocosms with four replicates of each dung type. Mesocosms consisted of 2.5 L opaque polypropylene containers with eight 2-mm drainage holes in the bottom and a 2-mm aperture mesh lid. Each mesocosm was filled to a depth of 13.5 cm of silica sand that had been thoroughly wetted with 600 mL of water 24 h prior to the start of the experiment. Though *O. nuchicornis* can tunnel below this depth, approximately 80% of brood balls are buried within the top 13 cm (Table S3).

Prior to the experiment start, each dung treatment was removed from the freezer and thawed in a 4 °C refrigerator for 48 h. Once thawed, all collected dung of each type was placed in a bucket and thoroughly homogenized using a drill with an 8 cm diameter paint mixer attachment. Each mesocosm then had an 85 g ball of homogenized dung placed on the sand surface. A total of $N = 100$ *O. nuchicornis* beetles were used in this study. Directly following the dung placement, five randomly selected beetles (two males and three females) were placed in each mesocosm on 14 July 2023, allowing for the formation of breeding pairs and providing opportunity for male choice between female mates.

Mesocosms were observed daily. Following an initial 7-day period, any beetle observed walking or flying within the mesocosm not actively utilizing the dung was removed from the experiment with the date and sex recorded. Any beetles found within the first 8 weeks (14 July 2023–8 Sept. 2023) were recorded as F0 generation. Any beetles found after 8 September 2023 were assumed to be F1 generation (offspring) based on a range of 44–63 days for *O. nuchicornis* to complete development under the laboratory temperature 20.5 °C (*Floate et al., 2015*). All beetles removed from the mesocosms were stored at −80 °C until pronotal measurements were taken at the end of the experiment.

Pronotal width was used to estimate the body size of each beetle and as an indicator of individual fitness. Increased body size has been found to result in increased measures of beetle fitness in terms of fecundity and mating success (*Marín-Armijos, Chamba-Carrillo & Pedersen, 2023*), and in the quantity of dung that a beetle can bury (*Manning & Cutler, 2020*). We also recorded the proportion of major male and minor male beetles, which is also an indicator of dung quality and beetle fitness (*Hunt & Simmons, 2000*). Beetles were removed from the freezer and measured for pronotal width using a FineSource carbon fiber composite digital caliper. The widest part of the pronotum was measured; this measurement was repeated until the largest width was recorded in triplicate.

One week following the end of the F1 observation period (10 November 2023), mesocosms were dismantled and remaining dung was removed from the surface of the sand. The sand was sifted through a 1 cm aperture sieve and examined for the presence of deceased beetles and evidence of nesting through the presence of tunneling and brood balls. Any intact beetle found during this process was removed and recorded as an $F_1$ beetle, as F0 beetles observed during this time were highly decomposed.

## Data analysis

We used a series of linear models to compare the effect of dung type on multiple endpoints. To avoid pseudo-replication, when multiple beetles were being measured within a single mesocosm, average values were calculated for each mesocosm (time F0 spent feeding = median, F1 pronotal width = mean). Within the linear models, we used different error distributions based on the type of data, and whether model residuals indicated evidence of overdispersion. The final models had the following error structures: F0 feeding and nesting time (quasi-Poisson), total number of F1 offspring (quasi-Poisson), pronotal width of F1 generation (Gaussian), proportion of major males compared to total males (Binomial). We also tested the correlation of nutritional parameters and F1 beetle endpoints. All models

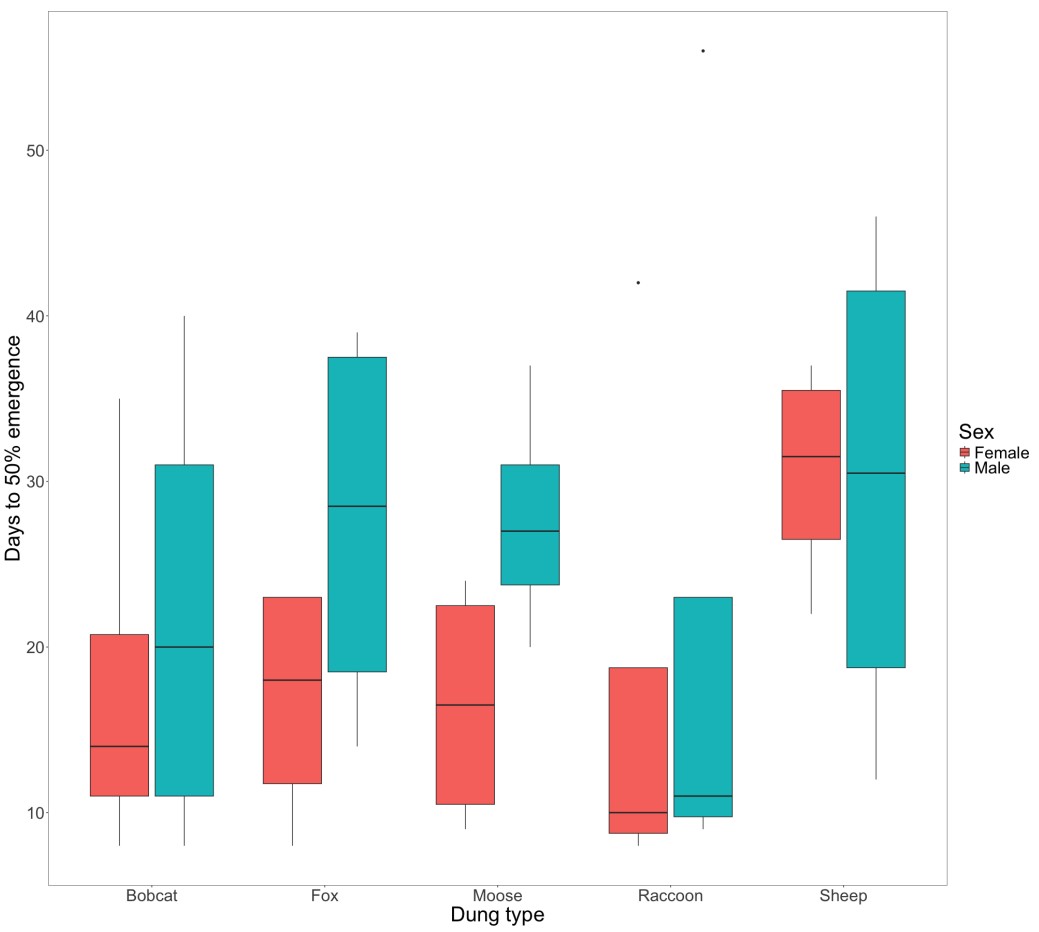

**Figure 2** **Box and whisker plot visualizing the median number of days since experiment start to 50% F0 beetle emergence for females and males.** The horizontal line inside each box represents the median time in days (female bobcat = 14, male bobcat = 20, female fox = 18, male fox = 28.5, female moose = 16.5, male moose = 27, female raccoon = 10, male raccoon = 11, female sheep = 31.5, male sheep = 30.5). The upper and lower boundaries of the coloured boxes show the inter-quartile range (IQR). The whiskers extending from the boxes show maximum and minimum values for data within 1.5x the IQR; any observations beyond this range are represented as a point.

were run using R Statistical Software (v4.2.2; *R Core Team, 2022*), and figures were made using ggplot2 (*Wickham, 2016*) and cowplot (*Wilke, 2024*).

## RESULTS

There was little evidence to suggest that dung type affected the amount of time that F0 beetles spent within the dung for either female ($X^2_{(4,20)} = 26.13$, $P = 0.391$) or male beetles (Fig. 2, $X^2_{(4,20)} = 8.36$, $P = 0.933$). Timing of F0 emergence ranged from 7 -56 days, though only 50% ($n = 10$) mesocosms had all 5 beetles re-emerge, with the remaining 10 mesocosms each missing a single beetle (presumed dead).

Evidence strongly suggested that the total number of successfully developed F1 beetles varied amongst treatments (Fig. 3, $X^2_{(4,20)} = 33.54$, $P = .0005$). Beetles using fox (2.25 ±

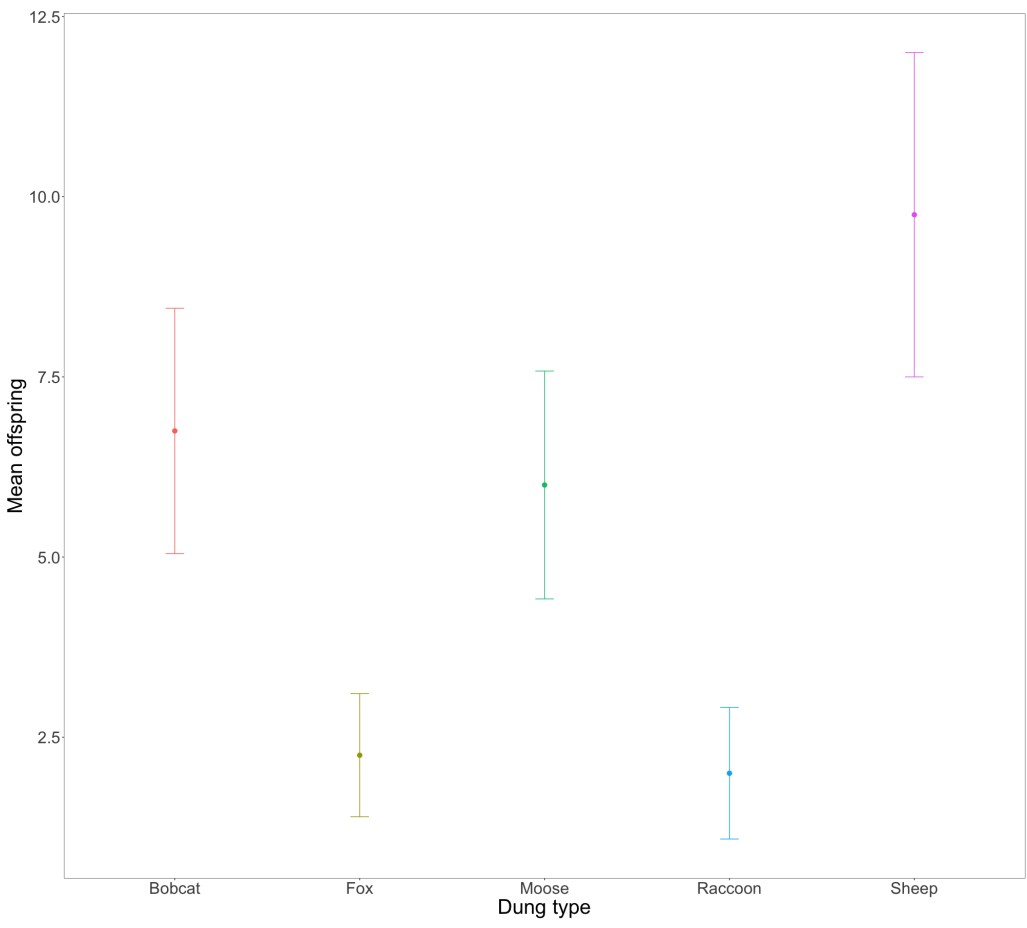

**Figure 3** **Mean number and standard error (SE) of F1 beetles that successfully reached full development across $n = 4$ replicates per dung treatment.** F0 beetles using fox ($2.25 \pm 0.85$) and racoon dung ($2 \pm 0.91$), produced significantly fewer F1s than beetles using sheep ($9.75 \pm 2.25$), moose ($6 \pm 1.58$), or bobcat ($6.75 \pm 1.70$) dung.

0.85) and raccoon dung ($2 \pm 0.91$) produced significantly fewer offspring per mesocosm than beetles using bobcat ($6.75 \pm 1.70$), moose ($6 \pm 1.58$), and sheep dung ($9.75 \pm 2.25$). Beetles from two mesocosms did not produce any offspring –1 with racoon dung and 1 with fox dung. The greatest number of offspring produced by any one mesocosm was 15 (sheep dung).

We found little evidence to suggest that dung type affected pronotal width in females ($F(4) = [2.85]$, $P = [0.071]$); major males: ($F(4) = [1.17]$, $P = [0.381]$); or minor males: ($F(3) = [0.589]$, $P = [0.648]$) from the F1 generation (Fig. 4).

A comparison of the proportion of male to female F1s as a function of treatment revealed no evidence to suggest a differing effect ($F(4,14) = [1.639]$, $P = [.224]$) (Fig. 5A). The consistent ratio of female to male developed offspring across treatments indicates that sex does not affect the likelihood of a beetle completing development across different dung types.

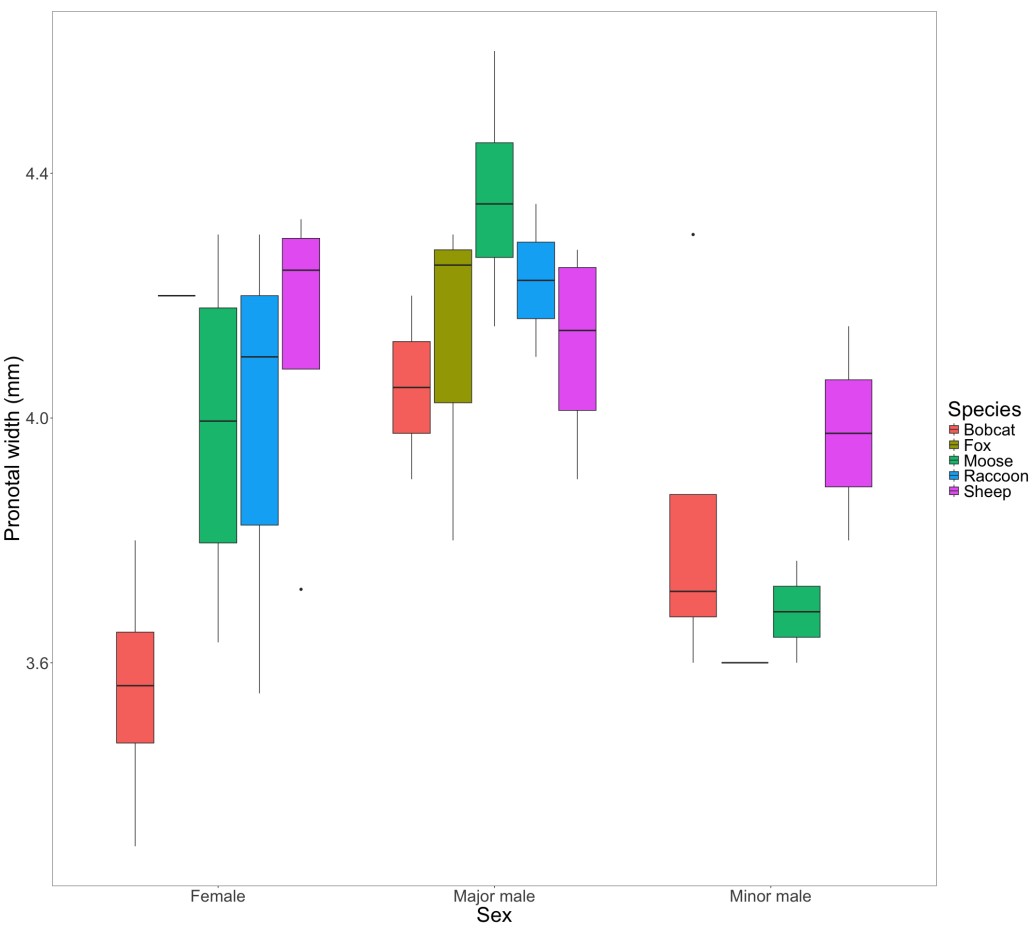

**Figure 4** **Box and whisker plot visualizing the mean pronotal width for female, major male, and minor male F1 beetles across dung treatments.** The horizontal line inside each box represents the median. The upper and lower boundaries of the coloured boxes show the inter-quartile range (IQR). The whiskers extending from the boxes show maximum and minimum values for data within 1.5x the IQR; any observations beyond this range are represented as a point.

There is weak to moderate evidence to suggest that the difference in the proportion of major male to total male F1s varied across dung treatments (F(4,14) = [3.232], $P$ = [.051]) (Fig. 5B). Beetles using bobcat dung produced the lowest proportion of major males (0.19 ± 0.24 SD), while beetles using fox, moose, and sheep dung produced primarily major males (0.83 ± 0.29; 0.65 ± 0.41; 1 ± 0; 0.75 ± 0.32).

## DISCUSSION

One of the primary concerns we address in this study is the risk of *O. nuchicornis* spreading beyond agroecosystems to use the dung of wild animals. Providing sheep dung as a treatment provided a baseline estimate as to what reproduction looks like with a preferred resource. We found that while *Onthophagus nuchicornis* performs best using sheep dung, this non-indigenous species can effectively use the dung of many types of animals indigenous

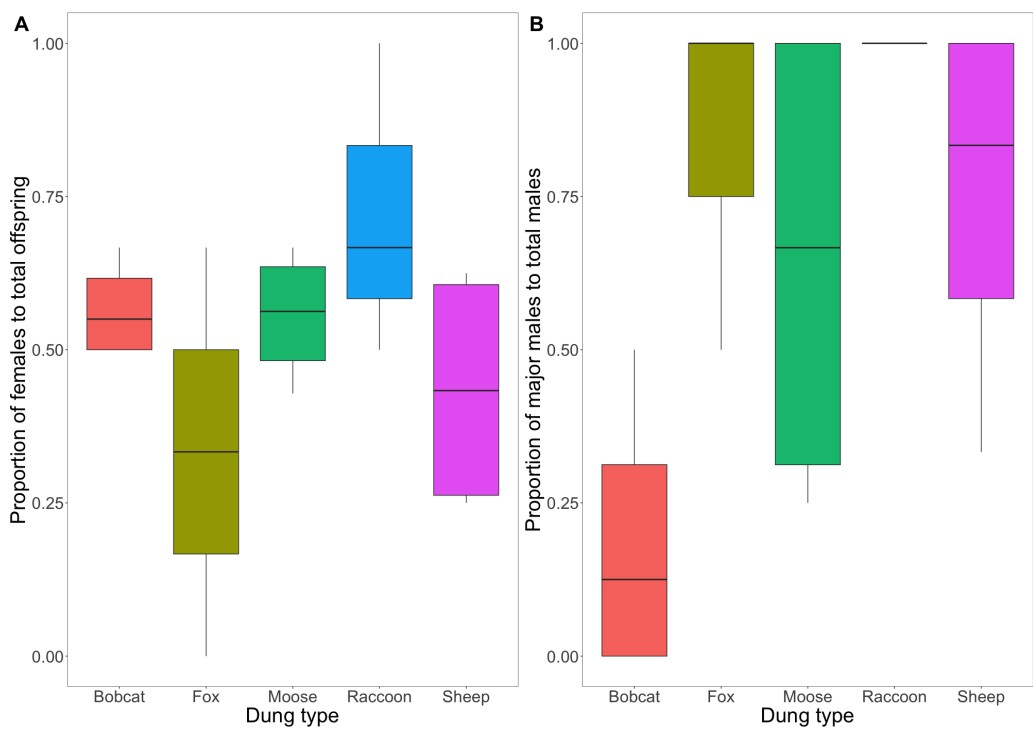

**Figure 5** **(A) Box and whisker plot visualizing the proportion of emerged F1 beetles that were female. (B) Box and whisker plot visualizing the mean proportion of major males to total males across dung treatments.** A value of 1 indicates 100% of male F1s produced were major males. The horizontal line inside each box represents the mean (A: bobcat = 0.57, fox = 0.33, moose = 0.56, raccoon = 0.72, sheep = 0.44; B: bobcat = 0.19, fox = 0.83, moose = 0.65, raccoon = 1, sheep = 0.75). The upper and lower boundaries of the coloured boxes show the inter-quartile range (IQR). The whiskers extending from the boxes show maximum and minimum values for data within 1.5x the IQR; any observations beyond this range are represented as a point.

to North America. The parameters we measured to evaluate dung suitability (F0 nesting time, offspring abundance, pronotal width, and proportion of major males) are widely used indicators of beetle fitness and resource suitability (*e.g., Manning & Cutler, 2020; Heddle et al., 2023; Yap, Toh & Puniamoorthy, 2024*). Our experimental approach did not allow us to assess how the fitness of the F1 beetles influenced their ability to disperse, overwinter, mate, and nest, all of which are essential to the production of the next generation of offspring.

All the beetles collected for use within this experiment were collected from sheep dung. Higher success in brood ball formation and breeding using sheep dung in no-choice experiments may reflect that sheep dung is a more suitable food and nesting substrate. In choice tests in both field (*Dormont, Epinat & Lumaret, 2004*) and laboratory settings (*Raine et al., 2019*), dung beetles show some degree of feeding preference –often showing lower attraction to the dung of carnivores relative to herbivores (*e.g., Martín-Piera & Lobo, 1996*). In Western Europe, where *O. nuchicornis* is native, it is most encountered in the dung of sheep, horse, and cow (*Biström, Silfverberg & Rutanen, 1991*), but has also been found using dog faeces (*Watkins & Mann, 2018*).
Higher levels of offspring emergence in sheep dung relative to other dung types may be evidence of local adaptation to sheep dung (*e.g.*, familiarity with resource use or presence of suitable bacterial symbionts in the gut). Rather than the beetles showing an affinity toward the dung from which they were most familiar (based on where they were collected), observed differences may reflect the lower suitability of alternative resources—due to differences in physio-chemical characteristics, such as low moisture content (*Frank et al., 2017*) or presence of aversive olfactory cues like phenol (*Mansourian et al., 2016*). *Hunt & Simmons (2004)* showed that maternal investment in the congeneric *O. taurus* depends on nutrition. When female beetles were given nutrient-rich horse dung and nutrient-poor cow dung, they compensated for the poorer cow dung by making larger brood balls. However, this led to fewer brood balls (and ultimately offspring) in cow dung compared to horse dung. Using a multi-generational evolutionary approach to resource adaptation could be a useful experimental approach in determining whether local adaptation underpins the differences we observed in this study.

The logistical challenge of working with native mammal dung has been relatively common in past studies. For example, in *Raine et al. (2018)* and *Raine et al. (2019)*, replication for dung types in beetle reproduction studies was as low as 1 for some species, based on constraints of dung availability. The number of replicates we were able to perform in this study were similarly limited by the amount of fresh dung that could be collected from the native mammals, resulting in low statistical power throughout our results. Future studies would benefit from a higher degree of replication to ensure the detection of all true effects, which may be under-detected in our study.

### F0 generation nesting and emergence

There was little evidence to suggest a significant difference in F0 post-nesting emergence times among dung treatments for male or female beetles (Fig. 2). We observed that male beetles took longer to emerge than female beetles (Fig. 2) as well as a notable increase in days to emergence by female beetles using sheep dung when compared to female beetles in other treatments (Fig. 2). While longer larval development times have been linked to higher fitness in adult beetles (*Shafiei, Moczek & Nijhout, 2001*), the reasons for differences in emergence timing for the F0 generation are unknown. Given that we found sheep dung to be the most suitable, this additional time spent underground by F0 females using sheep dung may be due to increased numbers of brood balls formed or possible increased size of brood balls (as observed by *Yap, Toh & Puniamoorthy, 2024*), both of which could align with an increase in resulting F1 number and fitness (*Hunt & Simmons, 2000*). Time spent underground by F0 beetles could also be due to increased tunnelling activities, though the production of more extensive tunnel systems is less clearly linked to dung suitability or success of the F1 generation.

### Offspring fitness

The congeneric species *Onthophagus taurus* has been observed to respond to food deprivation during larval development by pupating prematurely and reaching the adult stage of development sooner than larvae with sufficient access to food (*Shafiei, Moczek*

*& Nijhout, 2001*). This shorter development time leads to smaller adult beetles (*Buzatto, Tomkins & Simmons, 2012*; *Emlen, 1997*), and an increase in minor males (*Emlen, 1997*; *Yap, Toh & Puniamoorthy, 2024*). We observed a similar developmental response in beetles feeding on bobcat dung. Although the results were not statistically significant, beetles in the bobcat treatment had smaller pronotal widths and were the only dung treatment to produce more minor males than major males (major males only accounted for 22% of male offspring) (Figs. 4 and 5B). Small size (when compared to the control) despite the beetles having access to the same quantity of dung as other treatments, suggests that bobcat dung may be a less suitable resource for *O. nuchicornis*, though still sufficient in quality to allow larvae to complete development (Fig. 3).

Mesocosms with fox and raccoon dung produced significantly fewer offspring (fox = 1.25; raccoon = 1.75) than the other treatments (mean offspring produced was between 4 and 8 per treatment), including bobcat (Fig. 3). The offspring that did develop using fox and raccoon dung contained the highest proportion of major males to total male offspring (Fig. 5B). While the number of offspring produced indicates poor dung suitability for larval development (Fig. 3), the high proportion of major males for beetles using fox and raccoon dung conversely indicates highly suitable dung resources (Fig. 5B). Attraction to fox and raccoon dung by *Onthophagus* species is evident in other studies: *Fincher, Stewart & Davis (1970)* found fox dung to be strongly attractive to *Onthophagus* spp.–more so than cow, sheep, and horse dung. Raccoon dung was also moderately attractive (still more preferred than sheep and horse), though this study did not evaluate the ability of the beetles to reproduce using these dung types.

Beetles feeding on sheep dung produced the highest number of offspring (Fig. 3). While the offspring that developed on sheep dung do not have the widest pronotal measurements taken, the range of pronotal measurements across both sexes is consistently higher overall than from other types of dung (Fig. 4). The resulting implication that sheep dung was the most suitable among the dung types offered is reasonable given that *O. nuchicornis* is often found feeding on sheep dung in its native range (*Biström, Silfverberg & Rutanen, 1991*). Higher suitability of dung types from the beetles' native range has also been demonstrated through increased reproductive success of *O. lecontei* using wild rabbit dung in Mexico (*Arellano et al., 2015*).

## Dung nutrition and preference

Evidence favouring preference among dung beetles towards dung from trophic guilds is inconsistent, with some reports that dung beetles have a higher attraction to herbivore dung than that of carnivores or omnivores (*Martín-Piera & Lobo, 1996*) and other reports that, while dung beetles show different levels of attraction to different types of dung, there is no correlation with feeding guild (*Frank et al., 2017*). Both studies referenced above used pitfall traps baited with fresh dung to attract beetles, but did not allow the beetles to come into contact with the dung. *Martín-Piera & Lobo (1996)* demonstrated that numerous species of dung beetles, including seven *Onthophagus* species, showed the strongest attraction to cow and horse dung, and little attraction to omnivore and carnivore dung. Lynx dung (closely related to bobcat, *Lynx rufus*) was tested and found to be barely colonized by dung beetles

in a choice test under field conditions. This study took place in Spain, and observed that while dung beetles in Europe tend to show a preference towards domesticated herbivores, dung beetles in tropical and North American regions show little preference differentiation among feeding guilds (*Martín-Piera & Lobo, 1996*).

A more recent study by *Frank et al. (2017)* in Germany showed no consistent preference between feeding guilds among 23 dung beetle species in field conditions. This study tested forest and grassland sites, and while *O. nuchicornis* was among the species recorded, very few were present among the 1,191 individual beetles captured. All 23 types of dung tested attracted dung beetles, with the highest beetle abundance found on lynx, wild boar, and sheep dung. These preferences were not correlated with the diet or feeding guild of the animal—in fact, the dung types that proved most attractive for beetles were representative of all three feeding guilds (lynx—carnivore, wild boar—omnivore, and sheep—herbivore). In contrast again, strong attraction to dung from omnivorous mammals (swine and opossum) has been demonstrated by *Onthophagus* species (primarily *O. hecate*), while the dung of herbivorous and carnivorous mammals was notably less attractive and similar to one another in attractiveness (*Fincher, Stewart & Davis, 1970*).

*Frank et al. (2017)* also showed that the resource preferences of various dung beetle species could not be linked with any nutritional parameters associated with the dung, determining that resource attraction was primarily affected by olfactory response to dung volatiles, and that volatiles do not serve as indicators of nutritional quality to dung beetles. A lack of correlation between dung suitability and nutritional parameters has also been shown in *O. lecontei*'s preference of rabbit dung over horse or goat (*Arellano et al., 2015*). Here, using a no-choice test, we investigated whether the nutritional parameters of different dung types could be predictive of its suitability as a resource for nesting and reproduction, or the resulting offspring fitness. None of the nutritional parameters we measured (dry matter percentage, crude protein content, nitrogen content, C:N ratio) were correlated with any of our measured offspring outcomes, indicating that the investigated nutritional properties did not meaningfully influence resource suitability (Table S2). In addition to olfactory volatile cues, some evidence also suggests that mineral content and electrical conductivity may impact the suitability of the dung, though not necessarily its attractiveness (*Kaur, Holley & Andrew, 2021*).

## Presence of macrochelid mites

Within five mesocosms (raccoon = 3, bobcat = 2) we observed large numbers of macrochelid mites (Mesostigmata: Macrochelidae) on the surface of the dung and sand. Two female F1 beetles from the raccoon treatment and 1 female F1 beetle from the bobcat treatment had heavy mite loads (>30 mites per individual) upon removal from the experiment. We cannot confirm how these mites entered the experiment, but none of the F0 beetles added to the experiment had conspicuous mite loads. Macrochelid mites are wingless microarthropods (*Keum et al., 2016*) found in abundance in carrion and animal dung (*Glida, Bertrand & Peyrusse, 2003*). Within these habitats, macrochelid mites feed upon the eggs and young larvae of flies, thereby contributing to biological control of some pest flies (*Glida, Bertrand & Peyrusse, 2003*; *Niogret, Lumaret & Bertran, 2010*). Fertilized

female macrochelid mites will attach to the bodies of dung beetles and other flying insects to be transported to a new dung pat where a new colony of macrochelid mites can establish (*Glida, Bertrand & Peyrusse, 2003*; *Niogret, Lumaret & Bertran, 2010*). This mite-beetle relationship is often considered commensal, though *Kotiaho & Simmons (2001)* found that experimentally adding macrochelid mites to minor males of the dung beetle *Onthophagus binodis* reduced the mean lifespan by 18% (82 days to 67 days). A heavy mite load could be an indicator or a consequence of low fitness, or simply reflect different environmental conditions (phoretic mites had colonized some dung types prior to dung collection, but not others). Exploring the interaction of dung beetles and their environment might be a useful study system to better understand the evolutionary costs of phoresy.

## CONCLUSIONS

As predicted, sheep dung proved to be the most suitable resource for *O. nuchicornis*, although *O. nuchicornis* was able to use the other four dung types (bobcat, fox, moose, raccoon) with varying degrees of success. The biological possibility for *O. nuchicornis* to spread beyond pastured systems is possible: evidence here shows that it can effectively use dung from at least some wild mammals indigenous to eastern North America. Furthermore, *O. nuchicornis* has demonstrated attraction to many dung types in field trials (*Fincher, Stewart & Davis, 1970*; *Biström, Silfverberg & Rutanen, 1991*; *Floate, 2023*). Competition-based experiments and validating the fitness of F1 individuals that developed in different dung types under field conditions would be useful next steps in understanding the potential consequences of *O. nuchicornis* in the wider environment, but given the high suitability of sheep dung as a resource, there may not be a significant reason for *O. nuchicornis* to emigrate from grazed pastures where dung is plentiful to compete for the dung of wild animals that is less suited for beetle development, and is almost inevitably scarcer in space, abundance, and time.

## ACKNOWLEDGEMENTS

We would like to thank Samantha Bennett for helpful discussions and data collection. We also thank the staff at Shubenacadie Wildlife Park and the Dalhousie University Faculty of Agriculture Ruminant Animal Centre for their facilitation of, and assistance in dung collection.

### Funding

This research received funding from Natural Sciences and Engineering Research Council of Canada Discovery Grant RGPIN/03941-2022 (PM), a Sobey Agricultural Undergraduate Research Award (AI), and support from the Nova Scotia Federation of Agriculture Living Labs (PM). There was no additional external funding received for this study. The funders had no role in study design, data collection and analysis, decision to publish, or preparation of the manuscript.

### Grant Disclosures

The following grant information was disclosed by the authors:

Natural Sciences and Engineering Research Council of Canada Discovery Grant: RGPIN/03941-2022.

A Sobey Agricultural Undergraduate Research Award (AI).

The Nova Scotia Federation of Agriculture Living Labs (PM).

### Competing Interests

The authors declare there are no competing interests.

### Author Contributions

- Alexe Indigo performed the experiments, analyzed the data, prepared figures and/or tables, authored or reviewed drafts of the article, and approved the final draft.
- Katelyn Stokes conceived and designed the experiments, performed the experiments, authored or reviewed drafts of the article, and approved the final draft.
- Olivia Burchell performed the experiments, authored or reviewed drafts of the article, and approved the final draft.
- Paul Manning performed the experiments, analyzed the data, prepared figures and/or tables, authored or reviewed drafts of the article, and approved the final draft.

### Field Study Permissions

The following information was supplied relating to field study approvals (i.e., approving body and any reference numbers):

Dung samples were collected at Shubenacadie Wildlife Park.

### Data Availability

The raw data is available in the Supplemental Files.

### Supplemental Information

Supplemental information for this article can be found online at http://dx.doi.org/10.7717/peerj.18674#supplemental-information.

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
