# Peer review of "The non-indigenous dung beetle (Onthophagus nuchicornis) can effectively reproduce using the dung of indigenous eastern North American mammals"

_PeerJ, doi:10.7717/peerj.18674_

## Round 0.1 · original submission · Major Revisions

Reviewer 1 raises very valid and serious concerns about the design and sample size of your experiment. As pointed out clearly, the number of replicates per dung type is 4, which is indeed very low, and there is a bias towards sheep dung. You need to at least discuss these points in your manuscript. Reviewer 2 has also pointed out the lack of substantial discussion on the nutritional profile and quality of the dung. Also, you need to discuss the native species composition to anchor your argument for this manuscript. I would be happy to receive a substantially revised version of the manuscript that addresses all the queries and concerns raised by the reviewers.

Reviewer 1 ·

Basic reporting

This manuscript was generally written well. The introduction needs to add some background information around the known impact of dung quality on F1 size. The discussion is very long and needs revision. The discussion is very long and does not incorporate a large diversity of references to compare results. Reduce the length of the discussion.

Experimental design

There are two major issues which I believe weaken the results and validity of this study. Firstly, the low replication (4 replicates) for each dung type is not satisfactory even though you’ve added more beetles. This weakens the statistics significantly. Secondly, there is a significant bias towards sheep dung in the experiments as beetles were collected from sheep dung, provided sheep dung prior to experiments, and sheep dung was a treatment.

Validity of the findings

There are two major issues which I believe weaken the results and validity of this study. Firstly, the low replication (4 replicates) for each dung type is not satisfactory even though you’ve added more beetles. This weakens the statistics significantly. Secondly, there is a significant bias towards sheep dung in the experiments as beetles were collected from sheep dung, provided sheep dung prior to experiments, and sheep dung was a treatment.

Additional comments

Please see attached PDF

Annotated reviews are not available for download in order to protect the identity of reviewers who chose to remain anonymous.

Reviewer 2 ·

Basic reporting

I enjoyed reading this manuscript assessing the use of different dung types by an introduced dung beetle in Canada on captively bred vertebrate dung. It is a fairly straight-forward assessment of a single dung beetle species’ ability to grow and reproduce on the dung of five different species: one ubiquitous livestock species (sheep) and four native species.
I have a few queries that need to be addressed moving forward.
Abstract – the second sentence of the abstract (line 26) bring the concept of competition into focus – thus I was expecting to see some aspects of competition being assessed in the paper – I was surprised to read that the assessment of dung beetle reproduction showed no assessment of competition between species, or competition for different resource amount. I think the focus on competition in the abstract and the introduction should be removed, to reduce the expectations that competition will be assessed in this manuscript.

Experimental design

The experiments and assessments are set up in a straightforward and clear manner

The nutritional profile and nutritional quality of the dung samples were mentioned in the abstract (lines 34 and 35) but these were not deeply assessed in the manuscript. Data is presented in Table 2, but there does not seem to be an analysis of mentioned in the methods or presented in the results. This information really should be highlighted more to strengthen the relevance of the manuscript.

Validity of the findings

You also talk extensively about the lack of statistical significance – it does draw out that there seems to not be much happening with the dung beetles. I suggest that rather than saying that there was no significant difference throughout the manuscript, you use the language of evidence, and mention the biological differences in the data, rather than the statistical ones – see: Muff, S., Nilsen, E.B., O’Hara, R.B. & Nater, C.R. (2022) Rewriting results sections in the language of evidence. Trends in Ecology & Evolution, 37, 203-210.

Additional comments

Line 95 – here you set the reader up to presume that you will be some sort of competition experiment, or some interaction that the dung beetles will have competing for a dung resource – this should be removed from the intro and put into the discussion as something to do next.

Line 104/105 - you need to have some specific objectives/ questions here – at the moment it is very unclear what you will be assessing in the manuscript – this is also where I was still thinking that you would be assessing dung beetle competition for resources.

Line 366 – Section On Dung Nutrition and Preference - this section is not directly relevant to the study you carried out. The first two paragraphs do not relate back to any data in the study. On line 395 you mention that
‘we investigated whether the nutritional parameters of different dung types could be predictive of its suitability as a resource for nesting and reproduction, or the resulting offspring fitness.’
‘None of the nutritional parameters we tested (dry matter percentage, crude protein content, nitrogen content, C:N ratio) were correlated with any of our measured offspring fitness outcomes’

But I do not see any evidence of this in your results. No explicit correlation of nutritional parameters of dung was identified in the methods (ie in the Datra Analysis section starting on line 201) or in the results. But you say there was no relationship in the discussion.

Line 434 – I think the conclusions should end with a more explicit assessment of the importance of the findings of this study, rather than saying a competition-based experiment should be carried out - the competition-based expt should be embedded into the main discussion.

Minor issues –
Line 93 – species name mis-spelled

Line 340 – how much less time? How many more minor males than majors?
Line 347 – how many fewer offspring produced?
Line 355 – how long did it take to complete development?

Reviewer 3 ·

Basic reporting

.

Experimental design

.

Validity of the findings

.

Additional comments

The text was clearly written, without ambiguity and in correct English.
All requirements were met.
I made some comments in the text: lines 81, 93, 105, 121, 408 and 421.
In the references, please see line 466.

Annotated reviews are not available for download in order to protect the identity of reviewers who chose to remain anonymous.

---

## Round 0.2 · Minor Revisions

Thank you for submitting the revised manuscript. I request you to revise your current manuscript as per the inputs of Reviewer 2, so that I can recommend it for acceptance.

Reviewer 1 ·

Basic reporting

Excellent work. I am satisfied with the manuscript in its current state

Experimental design

Good

Validity of the findings

Good

Additional comments

I am happy with the amendments that the authors have made.

Reviewer 2 ·

Basic reporting

Overall fine - many of the discussion chapters need to have better coverage of the literature. Some paragraphs have 0 or only 1 reference. It would be good to see a better relationship between the results and the literature in the discussion paragraphs. Even when talking about low replication in your study - put this in context of other dung beetle studies dong similar experiments. There is a huge literature on dung beetles, so you should be able to get a good and relevant coverage of information to back up your claims or relate your findings to.

More references needed for paragraphs starting on these lines:
291
312
322
328
342
354
362
384
393

Experimental design

this has been dealt with in the updated manuscript I feel

Validity of the findings

this has been dealt with in the updated manuscript I feel

Additional comments

none - this is a nice expt. Understanding how introduced invasive dung beetles interact with native species is crucial to assess

---

## Round 0.3 · accepted · Accept

I am happy to accept your manuscript.